# Noble Gas—Silicon Cations: Theoretical Insights into the Nature of the Bond

**DOI:** 10.3390/molecules27144592

**Published:** 2022-07-19

**Authors:** Stefano Borocci, Felice Grandinetti, Nico Sanna

**Affiliations:** 1Dipartimento per la Innovazione nei Sistemi Biologici, Agroalimentari e Forestali (DIBAF), Università della Tuscia, L.go dell’Università, s.n.c., 01100 Viterbo, Italy; borocci@unitus.it (S.B.); n.sanna@unitus.it (N.S.); 2Istituto per i Sistemi Biologici del CNR, Via Salaria, Km 29.500, 00015 Monterotondo, Italy; 3Istituto per la Scienza e Tecnologia dei Plasmi del CNR (ISTP), Via Amendola 122/D, 70126 Bari, Italy

**Keywords:** bonding analysis, electron energy density, noble gas-silicon cations, noble gas complexes, noble gas inserted compounds

## Abstract

The structure, stability, and bonding situation of some exemplary noble gas-silicon cations were investigated at the MP2/aVTZ level of theory. The explored species include the mono-coordinated NgSiX_3_^+^ (Ng = He-Rn; X = H, F, Cl) and NgSiF_2_^2+^ (Ng = He-Rn), the di-coordinated Ar_2_SiX_3_^+^ (X = H, F, Cl), and the “inserted” FNgSiF_2_^+^ (Ng = Kr, Xe, Rn). The bonding analysis was accomplished by the method that we recently proposed to assay the bonding situation of noblegas compounds. The Ng-Si bonds are generally tight and feature a partial contribution of covalency. In the NgSiX_3_^+^, the degree of the Ng-Si interaction mirrors the trends of two factors, namely the polarizability of Ng that increases when going from Ng = He to Ng = Rn, and the Lewis acidity of SiX_3_^+^ that decreases in the order SiF_3_^+^ > SiH_3_^+^ > SiCl_3_^+^. For the HeSiX_3_^+^, it was also possible to catch peculiar effects referable to the small size of He. When going from the NgSiF_3_^+^ to the NgSiF_2_^2+^, the increased charge on Si promotes an appreciable increase inthe Ng-Si interaction, which becomes truly covalent for the heaviest Ng. The strength of the bond also increases when going from the NgSiF_3_^+^ to the “inserted” FNgSiF_2_^+^, likely due to the cooperative effect of the adjacent F atom. On the other hand, the ligation of a second Ar atom to ArSiX_3_^+^ (X = H, F, Cl), as to form Ar_2_(SiX_3_^+^), produces a weakening of the bond. Our obtained data were compared with previous findings already available in the literature.

## 1. Introduction

About 130 years after the discovery of argon [1], the chemistry of the noble gases seems to be a fascinating “saga” [2] wherein combative scientists never tire of challenging, and defeating, the proverbial inertness of the elements. The field currently embraces a rich synthetic chemistry of xenon and krypton [3,4,5,6,7] and countless species of helium, neon, argon, krypton, and xenon that are obtained in the gas phase [8,9], in liquid and supercritical fluids [10], in cold matrices [11], or at high pressures [12]. The binding partners include main-group or transition elements, and the bonding motifs range from the weakest non-covalent contacts to strong covalent bonds.

The capability of the noble gases, especially krypton and xenon, to combine with carbon is well established. The chemistry of synthesized xenon compounds is already rich [13,14], and numerous neutral species having Kr-C and Xe-C bonds were detected in cold matrices [11]. Over the years, interest was also extended to the interaction of noble gas (Ng) atoms with the heaviest elements of group XIV, particularly silicon. Neutral Ng-Si compounds are still experimentally elusive, even though there are theoretical predictions of species such as FXeSiF [15], FArSiF_3_ [16], FKrSiF_3_ [17], H*_n_*SiNgNSi (*n* = 1, 3; Ng = Xe, Rn) [18], and FNgSiY (Ng = Kr, Xe, Rn; Y = N, P) [19]. Experimental progress was made instead in the study of ionic species, particularly cationic. Noble gas–silicon cations are indeed of interest for several reasons. The complexes of Ng atoms with simple silylium ions, SiX_3_^+^, are prototypical systems to assay the activating interactions conceivably promoted by the more complex cationic silicon Lewis acids employed in catalysis [20]. The “tagging” of silicon cluster cations with Ng atoms is an effective mode to investigate the otherwise elusive structure and stability of these ionic intermediates [21], and the study of Ng-Si interactions is of prime interest to interpret the experimental findings [22]. Information about Ng-Si ionic species is also of interest in connection with the reactive ion etching of silicon by energetic Ng ions [23].

The first evidence about noble gas–silicon cations emerged in 1995, when Cipollini and one of us [24] reported the ligand-exchange reaction between Xe and protonated SiF_4_ so as to form gaseous XeSiF_3_^+^, a stable species with a Xe-Si bond. Working under different experimental conditions, Hopkinson, Bohme, and their coworkers [25] could subsequently obtain the three congeners ArSiF_3_^+^, KrSiF_3_^+^, and XeSiF_3_^+^ from the direct addition of Ng to SiF_3_^+^. They also gained evidence for the high-energy “inserted” isomers FXeSiF_2_^+^ and FKrSiF_2_^+^. Some years later, Roithová and Schröder [26] prepared gaseous NeSiF_2_^2+^ and ArSiF_2_^2+^ from the reaction between Ne and Ar and the superelectrophilic dication SiF_3_^2+^. The experimentally observed NgSiF_3_^+^ were theoretically investigated by Morino, Chattaraj, and their coworkers [27] as part of their extensive ab initio study on the structure, stability, and bonding character of NgSiX_3_^+^ (X = H, F, Cl, Br; Ng = He-Rn), Ng_2_SiH_3_^+^, and Ng_2_SiF_3_^+^. The structure and stability of the FArSiF_2_^+^, FKrSiF_2_^+^, FXeSiF_2_^+^, NeSiF_2_^2+^, and ArSiF_2_^2+^ were also assayed by density functional theory (DFT) calculations performed in conjunction with experimental studies [25,26]. The bonding situation of these species remained, however, unexplored, and the congeners FNgSiF_2_^+^ (Ng = He, Ne, Rn) and NgSiF_2_^2+^ (Ng = He, Kr, Xe, Rn) are still unreported. Taking into account this only limited available information even on experimentally observed species, it was the purpose of the present study to perform a comparative theoretical analysis of noble gas–silicon cations, with emphasis on the relationships between the various experimentally observed bonding motifs and the nature of Ng-Si bonds. In order to obtain strictly comparable data, we re-examined the NgSiX_3_^+^ (Ng = He-Rn; X = H, F, Cl) and Ar_2_SiX_3_^+^ (X = H, F, Cl) explored in the previous study [27] and extended the investigation to the still-unexplored NgSiF_2_^2+^ (Ng = He-Rn) and FNgSiF_2_^+^ (Ng = Kr, Xe, Rn). The analysis was accomplished by the method that we recently proposed [28,29,30,31] to assay the bonding situation of noble gas compounds. The obtained results were also compared with previous findings reported in the literature [24,25,26,27].

The paper is organized as follows. Section 2 and Section 3 give a brief account of the method of bonding analysis and the relevant computational details. Section 4 presents the obtained results, discussed family by family so as to best highlight analogies and differences between the various bonding situations. Some concluding remarks are given in Section 5.

## 2. Method of Bonding Analysis

Our method of bonding analysis [28,29,30,31] relies on the study of three functions, namely the electron density *ρ*(***r***) [32], the electron energy density *H*(***r***) [28,33,34], and the reduced density gradient (RDG) *s*(***r***) [35,36]. Any Ng-X bond (X = binding partner) is, in particular, assigned following the step-by-step procedure [31] briefly describedbelow. Further details are given in Refs. [28,29,30,31].

Step 1. Ng-X contact is ascertained by analyzing the *ρ*(*r*) and locating the corresponding bond path (BP) and bond critical point (BCP) (the classical AIM analysis).

Step 2.The topological analysis of the *H*(*r*) of the whole molecule is accomplished. This typically produces various critical points (HCPs) of rank 3 and signature −3, −1, +1, or +3. The contour lines these points belong to are collected as the HCP lines.

Step 3. The HCP lines are combined with a set of standard (STD) *H*(*r*) lines, a recommended choice being the patterns ±*k* × 10*^n^* (*k* = 0, 1, 2, 4, 8; *n* = −5 ÷ 6).

Step 4. The HCP/STD lines are plotted as 2D or 3D graphs, and the visual inspection of these graphs allows the assignment of the bond as type A, B, or C. As discussed in our previous studies [28,29,30,31], *H*(***r***) generally partitions the atomic space in two well recognizable regions, namely an inner one of negative values, indicated *H*^−^(***r***), and an outer one of positive values, indicated *H*^+^(***r***). The boundary of these regions falls at distance *R*^−^, which is typical of each atom; at this distance, *H*(***r*** = *R*^−^) = 0. When two atoms form a chemical bond, their *H*^−^(***r***) and *H*^+^(***r***) regions combine in modes that signal the nature of the interaction. Particularly for Ng-X bonds, it is possible to recognize three major situations. In *interactions of type A*, the atoms overlap all of the contour lines of their *H*^+^(***r***) regions and part of the contour lines of their inner *H*^−^(***r***) regions, the bond appearing as a continuous region of negative values of *H*(***r***)and plunging in a zone of positive values. The bond is topologically signed by a (3,+1) HCP falling on the bond axis. Typical examples are covalent bonds, or donor–acceptor interactions with some degree of electron sharing. In *interactions of type B*, the *H*^−^(***r***) region of Ng, again, overlaps with the *H*^−^(***r***) region of the binding partner, but (*i*) no HCP exists on the bond axis, and (*ii*) the Ng-X inter-nuclear region includes a (more or less wide) region of positive *H*(***r***). Typical examples are the complexes of Ng donors with strongly electropositive Lewis acceptors. In *interactions of type C*, the Ng and the binding partner overlap only part of their *H*^+^(***r***) regions, their *H*^−^(***r***) regions remaining perfectly closed and separated by a (more or less wide) region of positive *H*(***r***). The bond thus appears as two clearly distinguishable *H*^−^(***r***) regions, separated by a region of positive values of *H*(***r***). Typical examples are noncovalent contacts of variable nature.

**Step 5.** The assignment of the bond as being type A, B, or C is further refined by examining the *H*(***r***) along the Ng-X BP, particularly at around the BCP. This serves to confirm the nature of interactions of type A and to distinguish the interactions of type B and C into B-loose (B^l^) or B-tight (B^t^) and C-loose (C^l^) or C-tight (C^t^). The adopted criteria are given in Table 1.

**Step 6**. Once designated being type A, B^l^/B^t^, or C^l^/C^t^, the Ng-X bond is assayed in terms of the contribution of covalency. This is accomplished by integrating the *ρ*(***r***) and the *H*(***r***) over the volume *Ω_s_* enclosed by the *s*(***r***) isosurface associated with the Ng-X BCP. The value of the *s*(***r***) is chosen by examining, particularly at around the BCP, the *s*(***r***) vs. sign(λ_2_) × *ρ*(***r***) 2D plot [λ_2_ is the second eigenvalue of the Hessian matrix of *ρ*(***r***), with λ_1_ < λ_2_ < λ_3_]. The selected value of *s*(***r***) is the highest one that still avoids the contribution of the tails of the atomic densities. Typical values range between 0.2 and 0.5. Relevant quantities calculated over *Ω_s_* include the average value of *ρ*(***r***), *ρ_s_*(ave) and the average, maximum, and minimum values of *H*(***r***), *H*_s_(ave,max,min). Based on the values of these quantities, and on the sign of *H*(***r***) over *Ω_s_*, *H*(*Ω_s_*), the bond is designated covalent (Cov), partially covalent (pCov), or noncovalent (nCov) according to the criteria listed in Table 2.

**Step 7**. The bond is finally assigned using the notations Cov(Type), pCov[Type/*H*(*Ω_s_*)] or nCov(Type), for example Cov(A), pCov(B^t^/H^−/+^), or nCov(C^l^).

Additional indices. In developing the method, we found it convenient to introduce two additional numerical indices that allow the further assay of the degree of the various interactions. Thus, for interactions of type C, we defined [28,29,37] the degree of polarization of Ng, DoP(Ng), as the dimensionless index given by the equation
(1)$$\mathrm{DoP}\left(\mathrm{Ng}\right)=\frac{[R_{\mathrm{Ng}}^{-}\left(\mathrm{Ng}-X\right)-R_{\mathrm{Ng}}^{-}]\times 100}{R_{\mathrm{Ng}}^{-}}$$
where $R_{\mathrm{Ng}}^{-}\left(\mathrm{Ng}-X\right)$ is the radius of the *H*^−^(***r***) region of Ng along the axis formed by Ng and the Ng-X BCP, and $R_{\mathrm{Ng}}^{-}$ is the radius of the *H*^−^(***r***) region of the free atom. The DoP(Ng) measures, in essence, the deformation of the *H*^−^(***r***) region of Ng arising from the interaction with X. Its positive/negative sign signals Ng atoms polarized toward/opposite to X, and its magnitude is related to the extent of the polarization. For interactions of type A, borrowing a concept introduced so far by Espinosa et al. [38], we defined [28] the bond degree (BD) as the minus ratio between the *H*(***r***) and the *ρ*(***r***) calculated at the HCP: BD = −*H*(HCP)/*ρ*(HCP). We introduce here the average BD over *Ω_s_*, BD_s_(ave), defined as the average over *Ω_s_* of the ratio *H*(***r***)/*ρ*(***r***). Formulated in this way, the index is applicable to any type of interaction (A, B, or C).

## 3. Computational Details

The calculations were performed at the Møller–Plesset level of theory truncated at the second order (MP2) [39], using Dunning’s correlation consistent aug-cc-pVTZ [40] basis set (denoted here as aVTZ). The Xe and Rn atoms were treated with the aVTZ-PP basis set developed in conjunction with the Stuttgart/Cologne small-core, scalar-relativistic effective core potentials (ECP-28 and ECP-60, respectively) [41]. The geometry optimizations and frequencies calculations were performed with Gaussian 09 (G09) (Revision D1) [42], and the analysis of the *ρ*(***r***), the *H*(***r***), and the *s*(***r***) was accomplished with Multiwfn (version 3.8.dev) [43], using the wfx files generated with G09. *ρ*(***r***) is defined by the equation [32]:(2)$$\rho \left(r\right)=\overset{}{\underset{i}{\sum }}\eta_{i}{\left|\varphi_{i}\left(r\right)\right|}^{2}$$
where $\eta_{i}$ is the occupation number of the natural orbital $\varphi_{i}$, in turn expanded as a linear combination of the basis functions.

*H*(***r***) [28,33,34] is the sum of the kinetic energy density *G*(***r***) and the potential energy density *V*(***r***):(3)H(r)=G(r)+V(r)

The presently employed definition [32,44] of *G*(***r***) is given by the equation:(4)$$G\left(r\right)=\frac{1}{2}\overset{NO}{\underset{i=1}{\sum }}\eta_{i}{\left|\nabla \varphi_{i}\left(r\right)\right|}^{2}$$
where the sum runs over all the occupied natural orbitals, $\varphi_{i}$, of occupation numbers $\eta_{i}$. *V*(***r***) is evaluated [32] from the local form of the virial theorem:(5)$$V\left(r\right)=\frac{1}{4}\nabla^{2}\rho \left(r\right)-2G\left(r\right)$$

The planar (2D) plots of the HCP/STD lines of *H*(***r***) (*vide supra*) were produced with the Multiwfn [43].

*s*(***r***) is defined by the equation [35,36]:(6)$$s\left(r\right)=\frac{\left|\nabla \rho \left(r\right)\right|}{2{\left(3\pi^{2}\right)}^{\frac{1}{3}}\times \rho {\left(r\right)}^{\frac{4}{3}}}$$
and the integration of *ρ*(***r***) or *H*(***r***) over the volume *Ω_s_* enclosed by a given *s*(***r***) is accomplished by producing an orthogonal grid of points that encloses the isosurface and applying the formula
(7)$$P\left(\Omega_{s}\right)=\underset{i\left(\mathrm{RDG}<s\right)}{\sum }P\left(r_{i}\right)d_{x}d_{y}d_{z},\;$$
where *P*(***r****_i_*) is the value of *ρ*(***r***) or *H*(***r***) at the grid point ***r****_i_*, and *d_x_*, *d_y_*, and *d_z_* are the grid step sizes in the *x*, *y*, and *z* directions, respectively. The used values are *d_x_* = *d_y_* = *d_z_* = 0.025 *a*_0_, and the summation is carried out on all grid points ***r****_i_* having RDG < *s*. All the calculations were performed using *s* = 0.3.

## 4. Results and Discussion

### 4.1. The MP2/aVTZ Predicted Data

The connectivities and point groups of the presently investigated ions are shown in Figure 1. Their geometries were optimized at the MP2/aVTZ level of theory, and harmonic frequencies calculations confirmed that all the located structures were true minima on the corresponding potential energy surface. The electronic dissociation energies with the loss of Ng, *D*_e_, of the NgSiX_3_^+^ (Ng = He-Rn; X = H, F, Cl), NgSiF_2_^2+^ (Ng = He-Rn), and Ar_2_SiX_3_^+^ (X = H, F, Cl) were computed as the difference between the MP2/aVTZ electronic energy of the complex and that of Ng and of the SiX_3_^+^/SiF_2_^2+^/ArSiX_3_^+^ cation at the geometry it takes in the complex. The basis set superposition error (BSSE) was corrected using the counterpoise (CP) method by Boys and Bernardi [45]. The bonding analysis was as well accomplished at the MP2/aVTZ level of theory. The discussed data, given in Table 3, include the Ng-Si distances, *R*(Ng-Si); the *D*_e_; and the values of *Ω_s_*, *N*(*Ω_s_*), *ρ*_s_(ave), *H_s_*(ave/max/min), and BD_s_(ave). The 2D plots of the *H*(***r***) are given in Figure 2, Figure 3, Figure 4, Figure 5 and Figure 6. These Figures are, indeed, of major interest in discussing the bonding situation of the investigated species. They allow, in fact, the classification of the bonds as being type A, B, or C (*vide supra*) and the catching of subtle features of the Ng-Si interactions by examining, in particular, the shape of the *H*(***r***) at around the Ng atom (*vide infra*). The full Cartesian coordinates and the results of the classical AIM analysis are also available as Appendix A.

The good accuracy of our predicted MP2/aVTZ data is supported by the following arguments. First, as shown in Appendix A, the *T*_1_ diagnostics [46] (the norm of the vector ***t***_1_ of the single-excitation amplitudes from a coupled-cluster calculation with the inclusion of single and double excitations, CCSD, divided by the square root of the number of correlated electrons *N*, $T_{1}=\sqrt{\frac{t_{1}\cdot t_{1}}{N}}$)of the investigated ions resulted invariably within the threshold of 0.02 used to establish the validity of a mono-determinantal method (such as the MP2) to describe a wave function. The good performance, in particular, of the MP2/aVTZ is suggested by the comparison with the results obtained by Morino, Chattaraj, and their coworkers [27] from the study of NgSiX_3_^+^ (X = H, F, Cl, Br; Ng = He-Rn), Ng_2_SiH_3_^+^, and Ng_2_SiF_3_^+^. They computed the geometries and stabilities of these complexes at both the MP2 and the CCSD(T) (CCSD with an estimate of connected triples) using the def2-TZVP and def2-QZVPPD basis sets. They found that, for both basis sets, the MP2 and the CCSD(T) delivered comparable results. However, the in-principle more accurate def2-QZVPPD furnished *D*_e_ and *R*(Ng-Si) are larger and smaller, respectively, than those predicted using the def2-TZVP. Thus, taking also into account computational costs, they performed most of the calculations, including the bonding analysis, at MP2/def2-QZVPPD. According to our experience, for a given level of theory, the aVTZ generally furnishes results comparable with those obtained with def2-QZVPPD. Consistent with this expectation, we found that our MP2/aVTZ data are in very good agreement with the previous MP2/def2-QZVPPD estimates [27]. Thus, for NgSiX_3_^+^ (Ng = He-Rn; X = H, F, Cl) and Ar_2_SiX_3_^+^ (X = H, F), the two sets of values of the *R*(Ng-Si) feature a mean unsigned deviation (MUD) of 0.031 Å, and, for NgSiX_3_^+^ (Ng = He-Rn; X = H, F, Cl), the values of *D*_e_ (arriving up to ca. 45 kcal mol^−1^) feature a MUD of only 1.2 kcal mol^−1^. The AIM indices of NgSiH_3_^+^ (Ng = He-Rn) and Ar_2_SiH_3_^+^also unraveled quite similarly. The good accuracy of our MP2/aVTZ bonding analysis is also expected based on the extensive test calculations performed in our previous study [30], showing that this computational level furnishes results strictly similar to those obtained at the benchmark CCSD/aVTZ.

### 4.2. The NgSiX_3_^+^ (Ng = He-Rn; X = H, F, Cl): The Ng-Si Bond in Mono-Coordinated SinglyCharged Complexes

The ligation of any Ng to the Si atom of the singlet ground state SiX_3_^+^ (X = H, F, Cl) produces NgSiX_3_^+^ mono-coordinated structures with C_3v_ symmetry (see Figure 1). The values of the *R*(Ng-Si) of NgSiH_3_^+^ and NgSiF_3_^+^ progressively increase when going from Ng = He to Ng = Rn, ranging between ca. 2.12 and 2.74 Å and ca. 2.06 and 2.65 Å, respectively. On the other hand, the *R*(Ng-Si) of NgSiCl_3_^+^decreases from ca. 2.99 to ca. 2.67 Å when going from Ng = He to Ng = Kr and then increases up to ca. 2.79 Å for Ng = Rn. In any case, for any X, the values of *N*(*Ω_s_*), *ρ*_s_(ave), and BD_s_(ave) invariably increase when going from the He to the Rn congener, and *H*_s_(ave) becomes progressively more negative in the same order. The values of the *D*_e_ follow the same trend, and we ascertained, in particular, positive correlations between BD_s_(ave) and *D*_e_, well-fitted (*r*^2^ > 0.99) by exponential equations of the form *D*_e_ = *A*·exp[*B*·BD_s_(ave)], with comparable *A*/*B* values of 1.898 kcal mol^−1^/6.964 *e* hartree^−1^ (X = H), 1.918 kcal mol^−1^/6.571 *e* hartree^−1^ (X = F), and 1.975 kcal mol^−1^/6.179 *e* hartree^−1^ (X = Cl). One also notes that, for any Ng, the values of *R*(Ng-Si), *D*_e_, and BD_s_(ave) follow invariably the same order, namely NgSiF_3_^+^ > NgSiH_3_^+^ > NgSiCl_3_^+^. These trends actually herald the different bonding situations occurring in the various complexes. All of these systems are, in fact, stabilized by donor–acceptor interactions between Ng and SiX_3_^+^, the nature of the ensuing Ng-Si bonds depending on the size and polarizability of Ng and on the Lewis acidity of SiX_3_^+^. We first examine NgSiH_3_^+^. In HeSiH_3_^+^ (Figure 2a), the small He penetrates so close to Si as to undergo an appreciable deformation of its *H*^−^(***r***) region, measured by a DoP(He) as high as 40.1. This polarization, however, is not sufficient to promote contact with the *H*^−^(***r***) region of SiH_3_^+^, and the bond is, therefore, of type C. The *H*(***r***) at around the BCP of the He-Si bond is, however, negative at both the He and the Si side, and its values over *Ω_s_* range from negative to positive, being slightly negative on the average. The bond is, thus, designated pCov(C^t^/H^−/+^). In the NeSiH_3_^+^, the *H*^−^(***r***) region of Ne looks (nearly) spherical (Figure 2b), and the DoP(Ne) amounts to only 6.42. As a matter of fact, with respect to He, the bigger Ne is located further away from Si (2.330 Å vs. 2.122 Å; see Table 3), and this produces an interaction of type C with a lower degree of polarization of Ng. The polarizability (α) of Ne (0.3956 Å^3^), is, however, sufficiently higher than that of He (0.2055 Å^3^) to promote quantitative effects that are higher than those occurring in HeSiH_3_^+^. Thus, for the Ne-Si bond, not only the *H*(***r***) at around the BCP is negative at both the Ne and the Si side, but its values over *Ω_s_* are also invariably negative. The interaction is thus designated pCov(C^t^/H^−^). One also notes from Table 3 that, when going from HeSiH_3_^+^ to NeSiH_3_^+^, *N*(*Ω_s_*) and *ρ*_s_(ave) increase, respectively, from 4.86 to 7.57 m*e*, and, from 0.0156 to 0.0168 *ea*_0_^−3^, *H*_s_(ave) decreases (becomes more negative) from −0.000075 to −0.0013 hartree *a*_0_^−3^, and BD_s_(ave) increases from 0.0042 to 0.0747 hartree *e*^−3^. The *D*_e_ also increases from 2.0 to 3.1 kcal mol^−1^, and this is consistent with the major stabilizing role of the polarization unraveled by the energy decomposition analysis performed by Morino, Chattaraj, and their coworkers [27].

The change in the bonding situation of NgSiH_3_^+^ is even more dramatic when going from NeSiH_3_^+^ to ArSiH_3_^+^. The α of Ar (1.6411 Å^3^) is, in fact, sufficiently large to promote an extensive overlapping of its *H*^−^(***r***) region with that of SiH_3_^+^ (see Figure 2c),represented by a (3,+1) HCP along the Ar-Si axis. *H*(***r***) is also invariably negative over the *Ω_s_*, and the interaction is, therefore, of the A/H^−^ type. Not unexpectedly, the same character is assigned to the Ng-Si bonds occurring in the heaviest congeners KrSiH_3_^+^, XeSiH_3_^+^, and RnSiH_3_^+^, the α values of Kr, Xe, and Rn being, indeed, higher than that of Ar (2.4844, 4.044, and 5.3 Å^3^, respectively). The *ρ*_s_(ave) of these Ng-Si bonds, ranging between 0.0325 (Ar-Si) and 0.0421 *ea*_0_^−3^ (Rn-Si), is, however, well below the threshold of covalency (0.08 *ea*_0_^−3^) and all are thus designated pCov(A/H^−^).

Based on the criteria proposed by Boggs et al. [47], Morino, Chattaraj, and their coworkers [27] assigned the Ng-Si bonds occurring in all of the NgSiH_3_^+^ as possessing a covalent or partially covalent character. They noticed, however, that the geometries and AIM indices of HeSiH_3_^+^ and NeSiH_3_^+^ were most-suggestive of noncovalent interactions. As a matter of fact, our analysis confirms the assignment based on the Boggs criteria, all of the bonds occurring in NgSiH_3_^+^ featuring a contribution of covalency. This highlights the importance of comparing the results of different methods when assaying bonding situations that are at the borderline of different characters.

As discussed previously [27], the Lewis acidity of SiH_3_^+^, SiF_3_^+^, and SiCl_3_^+^ decreases in the order SiF_3_^+^ > SiH_3_^+^ > SiCl_3_^+^, and this trend is clearly recognizable in the nature of the Ng-Si bonds occurring in the various NgSiX_3_^+^ (X = H, F, Cl). Most illustrative in this regard are the three helium complexes. Thus, the comparison between Figure 2a and Figure 3a clearly shows that, when going from HeSiH_3_^+^ to HeSiF_3_^+^, the polarization of He is enhanced to such an extent that its *H*^−^(***r***) region comes into contact with the *H*^−^(***r***) region of SiF_3_^+^. The type of the He-Si bond thus changes from C to A, and the BD_s_(ave) increases by more than ten times, passing from 0.0042 to 0.043 hartree *e*^−3^. The *H*(***r***) over *Ω_s_*, however, is still partially positive, and the bond is overall designated pCov(A/H^−/+^). On the other hand, in HeSiCl_3_^+^, the He atom is appreciably less polarized than in HeSiH_3_^+^. As shown in Figure 4a, its*H*^−^(***r***) region is nearly spherical, and the DoP(He) amounts to only 3.53. The *H*(***r***) is also positive at both sides of the BCP and is invariably positive over *Ω_s_*. The He-Si bond is thus designated nCov(C^l^), with a negative BD_s_(ave) of −0.272 hartree *e*^−1^. Likewise the Ne-Si bond of NeSiH_3_^+^, the Ne-Si bonds of both NeSiF_3_^+^ and NeSiCl_3_^+^ are of type C, as evinced from the graphs shown in Figure 3b and Figure 4b. The three Ne-Si bonds feature, however, differences again related to the Lewis acidity of the cation decreasing in the order SiF_3_^+^ > SiH_3_^+^ > SiCl_3_^+^. The contacts occurring in NeSiH_3_^+^ and NeSiF_3_^+^ are, in fact, both designated pCov(C^t^/H^−^), but the BD_s_(ave) are appreciably different and are predicted to be 0.0774 and 0.135 hartree *e*^−1^, respectively. The contact occurring in NeSiCl_3_^+^ is, instead, nCov(C^l^), with a BD_s_(ave) of −0.086 hartree *e*^−1^.

The acceptor ability of SiF_3_^+^/SiCl_3_^+^being higher/lower than that of SiH_3_^+^ also became apparent when examining the three argon complexes. As evinced from Figure 3c, the Ar-Si bond of ArSiF_3_^+^ is of type B, the Ar atom being polarized to such an extent that its *H*^−^(***r***) region comes (nearly) into contact with the *H*^+^(***r***) region of SiF_3_^+^. This is, indeed, typical of complexes of Ng donors with strong Lewis acceptors [30]. The interaction also features an appreciable contribution from covalency, which is overall designated pCov(B^t^/H^−^). On the other hand, likewise ArSiH_3_^+^, the Ar-Si bond of ArSiCl_3_^+^ is designated pCov(A/H^−^), but all of the bond indices appreciably decrease (see Table 3). We note, for example, the BD_s_(ave) passing from 0.283 to 0.157 hartree *e*^−1^. The Ng-Si bonds of the complexes of Kr, Xe, and Rn with SiF_3_^+^ (Figure 3d–f) and SiCl_3_^+^ (Figure 4d–f) are also designated pCov(A/H^−^), and their bond indices again follow the decreasing trend NgSiF_3_^+^ > NgSiH_3_^+^ > NgSiCl_3_^+^.

### 4.3. The Ar_2_(SiX_3_^+^) (X = H, F, Cl): The Ng-Si Bond in Di-Coordinated SinglyCharged Complexes

The results obtained by Morino, Chattaraj, and their coworkers [27] clearly uncovered that the ligation of a second Ng atom to any NgSiX_3_^+^ produces a weakening of the Ng-Si interaction. They found that, as the Ng-Si distances increase, the complexation energies per Ng atom decrease, and the indices employed within various methods of bonding analysis invariably suggested a decreased degree of covalency. To understand the information gained with our taken approach, we explored the three exemplary Ar_2_(SiX_3_^+^) (X = H, F, Cl). The comparison with the corresponding mono-coordinated ArSiX_3_^+^ confirmed the indications from the previous study [27]. The weakening of the Ar-Si bond produced by the ligation of a second Ar atom is particularly evident for the chlorine complexes. A comparison between Figure 4c and Figure 5c shows, in fact, that when going from ArSiCl_3_^+^ to Ar_2_SiCl_3_^+^, the type of the interaction changes from A to C and the overall assignment changes from pCov(A/H^−^) to pCov(C^t^/H^−^^/+^) (see Table 3). In essence, the polarization exerted by SiCl_3_^+^ when interacting with two Ar atoms is insufficient to promote the overlapping of the *H*^−^(***r***) regions. This reduces the role of covalency, and BD_s_(ave) also drastically declines from 0.157 to 0.0036 hartree *e*^−1^. As for Ar_2_SiH_3_^+^ and Ar_2_SiF_3_^+^, based on the graphs shown in Figure 5a,c and our adopted criteria of classification, their Ar-Si bonds are designated pCov(A/H^−^). All the bond indices, however, indicate that they are weaker than the bonds occurring in the mono-coordinated ArSiH_3_^+^ and ArSiF_3_^+^. We note, in particular, values of BD_s_(ave) decreasing, respectively, from 0.283 to 0.190 hartree *e*^−1^ and from 0.354 to 0.265 hartree *e*^−1^. The complexation energies per Ar atom also decrease, respectively, from 14.1 to 8.1 kcal mol^−1^ and from 21.7 to 10.6 kcal mol^−1^.

### 4.4. The NgSiF_2_^2+^ (Ng = He-Rn): The Ng-Si Bond in Mono-Coordinated DoublyCharged Complexes

As mentioned in the Introduction, the bonding situation of the six NgSiF_2_^2+^ (Ng = He-Rn) is still unexplored. The plots of their *H(**r**)* (Figure 6) clearly uncover the extensive polarization of Ng toward the strong Lewis acceptor SiF_2_^2+^ with the formation of peculiarly tight bonds. This is consistent with the short bond distances between 1.721 (Ng = He) and 2.518 Å (Ng = Rn) and the high complexation energies between 14.0 (Ng = He) and 137.4 kcal mol^−1^ (Ng = Rn). The BD_s_(ave) are also generally high and range between 0.155 (Ng = He) and 0.552 hartree *e*^−1^ (Ng = Kr).

The interactions occurring in HeSiF_2_^2+^ and NeSiF_2_^2+^ are both designated pCov(B^t^/H^−/+^), with rather high values of *ρ_s_*(ave) (0.0383 and 0.0434 *e a*_0_^−3^, respectively), and BD_s_(ave) (0.155 and 0.159 hartree *e*^−1^, respectively). The Ar-Si bond of ArSiF_2_^2+^ is designated pCov(B^t^/H^−^), but its *ρ_s_*(ave) of 0.0743 *e a*_0_^−3^ points to an incipient covalent bond. True covalent bonds are, indeed, predicted for KrSiF_2_^2+^ and XeSiF_2_^2+^, the occurring interactions being designated Cov(A). The bond occurring in RnSiF_2_^2+^ is pCov(A/H^−^), but its ρ_s_(ave) of 0.0786 *e a*_0_^−3^ suggests a situation at the border of covalency.

A comparison of the data obtained for the NgSiF_2_^2+^ and the NgSiF_3_^+^ (*vide supra*) clearly uncovers that the increased charge at the Si atom dramatically enhances the degree of the interaction with the noble gas, shifting the Ng-Si bonds towards the domain of covalency. In order to explore the character of these bonds when Ng is inserted into the Si-F bond of SiF_3_^+^, we performed a bonding analysis of the FNgSiF_2_^+^. The obtained results are discussed in the following paragraph.

### 4.5. The FNgSiF_2_^+^ (Ng = Kr, Xe, Rn): The Ng-Si Bond in Inserted Cations

We searched for the six FNgSiF_2_^+^ (Ng = He-Rn), but only the heaviest congeners FKrSiF_2_^+^, FXeSiF_2_^+^, and FRnSiF_2_^+^ were located as stationary points on the corresponding potential energy surface and characterized as true minima. The Ar congener FArSiF_2_^+^, located so far by Hopkinson, Bohme, and their coworkers [25] as an energy minimum at the DFT level of theory (B3LYP/DZVP), was not confirmed here at the MP2/aVTZ. This tendency of DFT methods to overestimate the stability of only marginally stable “inserted” noble gas compounds, especially those containing He, Ne, and Ar, is not surprising and is already documented in the literature [48,49].

The bonding situation of FNgSiF_2_^+^ (Ng = Kr, Xe, Rn), not explored in the previous study [25], clearly emerges by examining the plots shown in Figure 5d–f and the data quoted in Table 3. All the Ng-F and Ng-Si bonds are of type A, and, over Ω_s_, the *H*(***r***) is invariably negative. The *ρ_s_*(ave) of any Ng-F bond is also definitely higher than 0.08 *e a*_0_^−3^, and these interactions are safely designated Cov(A). The *ρ_s_*(ave) of the Ng-Si bonds are also rather high at around 0.07 *e a*_0_^−3^ but are still below the threshold of covalency; these interactions are, therefore, designated pCov(A/H^−^). In any case, the corresponding BD_s_(ave) of 0.487 hartree *e*^−1^ (Kr-Si), 0.464 hartree *e*^−1^ (Xe-Si), and 0.441 hartree *e*^−1^ (Rn-Si) are invariably higher than the values predicted for the Kr-Si, Xe-Si, and Rn-Si bonds of the monocoordinated KrSiF_3_^+^, XeSiF_3_^+^, and RnSiF_3_^+^ (0.431, 0.470, and 0.471hartree *e*^−1^, respectively). In essence, the insertion of Ng into the Si-F bond of SiF_3_^+^ produces Ng-Si bonds tighter than those occurring in the mono-coordinated NgSiF_3_^+^. This reflects the further stabilizing role of the electron-withdrawing F atom bound to Si.

## 5. Concluding Remarks

The purpose of the present study was to compare, using a uniform and accurate level of theory, the bonding situation of some exemplary noble gas-silicon cations. We re-examined, in particular, the previously reported NgSiX_3_^+^ and Ar_2_SiX_3_^+^ (X = H, F, Cl) [27] and extended the study to the still unexplored NgSiF_2_^2+^, and FNgSiF_2_^+^. It was thus possible to gather a comprehensive view of the nature of the bonds occurring in the various experimentally observed species [24,25,26] and in their still unreported congeners. Ng-Si bonds generally feature a contribution of covalency, arising from the strong polarization of Ng by the Si atom. The effect is quite extensive for the systems containing Ar, Kr, Xe, and Rn but is also appreciable for those containing He and Ne. The small size of He promotes, in particular, peculiar effects not observed for the Ne congeners. As for the singly charged NgSiX_3_^+^, in keeping with the results of a previous theoretical study [27], we found that, for any X, the degree of the interaction and the role of covalency generally increase when going from Ng = He to Ng = Rn. In addition, for any Ng, these two factors progressively decrease in the order NgSiF_3_^+^ > NgSiH_3_^+^ > NgSiCl_3_^+^, this trend strictly mirroring the Lewis acidity of the cation decreases in the same order. The comparison between the NgSiF_3_^+^ and NgSiF_2_^2+^ also uncovered the dramatic effect of the increased charge in enhancing the stability of the complexes and the degree of the interaction. The Ng-SiF_2_^2+^ bonds are, in fact, truly covalent in nature for Ng = Kr, Xe, and Rn. We also ascertained that the insertion of Kr, Xe, and Rn into the Si-F bond of SiF_3_^+^, so as to form FNgSiF_2_^+^, produces Ng-Si bonds appreciably tighter than those occurring in the corresponding NgSiF_3_^+^, being of incipient covalent character. We refer to this as the cooperative effect exerted by the adjacent F atom. Finally, the bonds occurring in the mono-coordinated xenon complexes invariably feature an appreciable contribution of covalency. This supports the conclusion reached in a previous study [22] of an incipient chemical bond between Xe and the cationic silicon cluster Si_4_^+^.

## Figures and Tables

**Figure 1 molecules-27-04592-f001:**
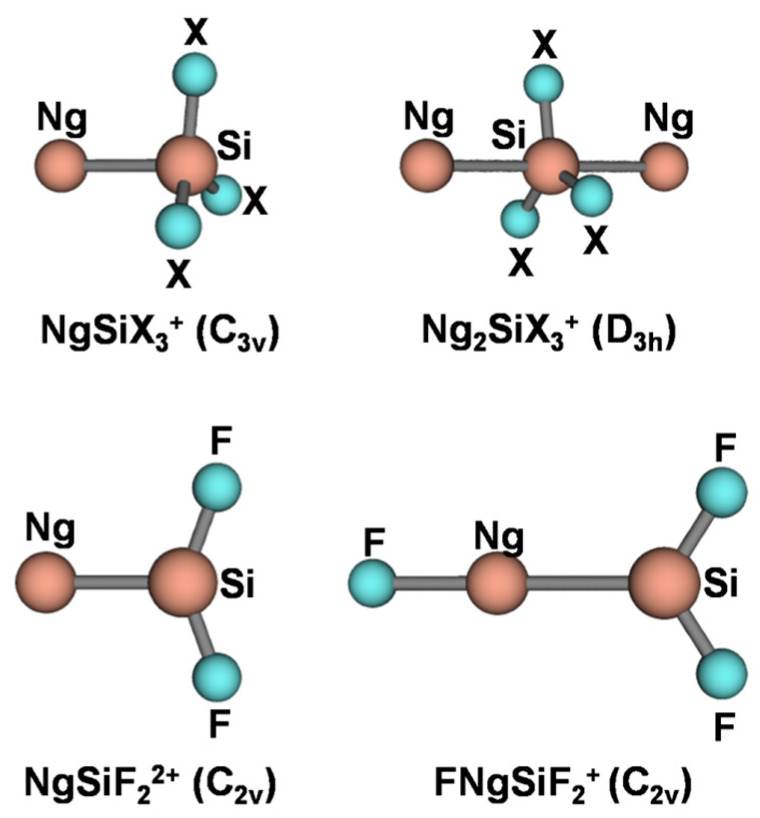
Connectivities and point groups of the Ng-Si cations.

**Figure 2 molecules-27-04592-f002:**
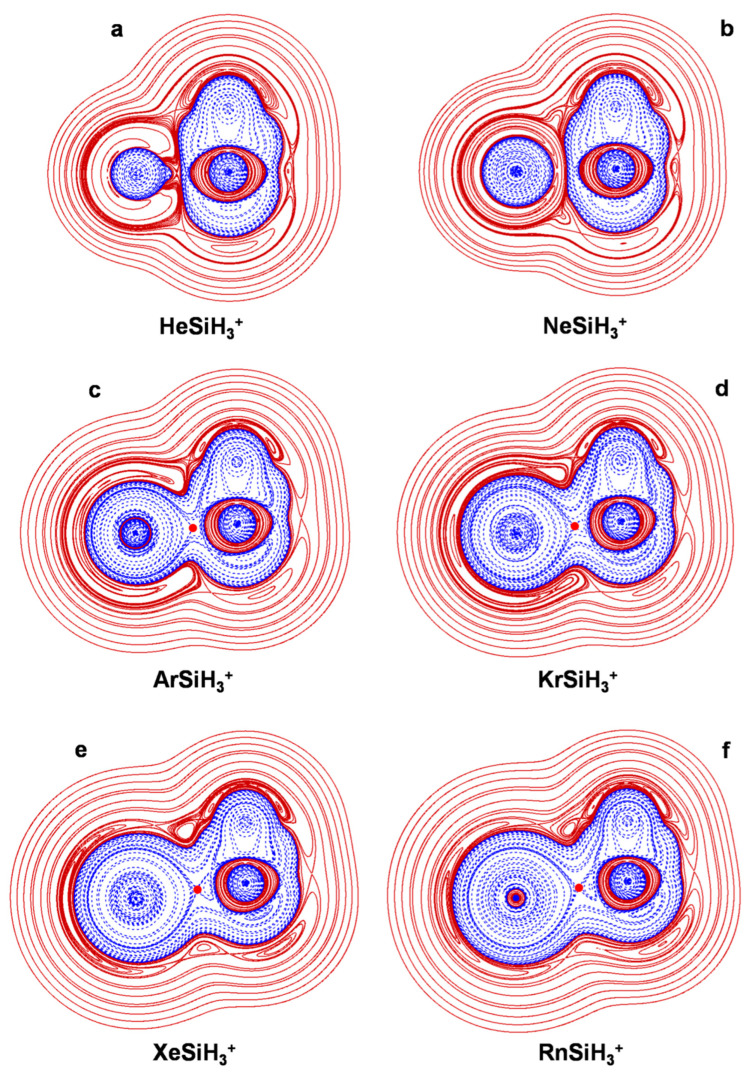
2D plots of *H*(***r***) in the symmetry plane of (**a**) HeSiH_3_^+^; (**b**) NeSiH_3_^+^; (**c**) ArSiH_3_^+^; (**d**) KrSiH_3_^+^; (**e**) XeSiH_3_^+^; (**f**) RnSiH_3_^+^ (solid/brown and dashed/blue lines correspond, respectively, to positive and negative values). The red dots represent the HCPs.

**Figure 3 molecules-27-04592-f003:**
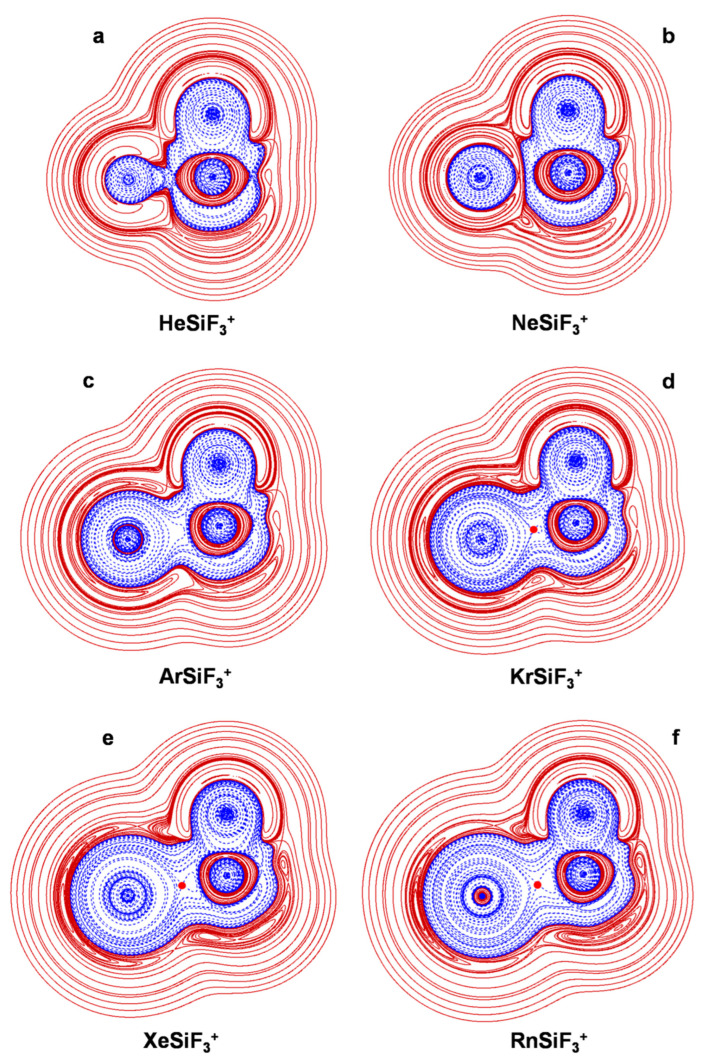
2D plots of *H*(***r***) in the symmetry plane of (**a**) HeSiF_3_^+^; (**b**) NeSiF_3_^+^; (**c**) ArSiF_3_^+^; (**d**) KrSiF_3_^+^; (**e**) XeSiF_3_^+^; (**f**) RnSiF_3_^+^ (solid/brown and dashed/blue lines correspond, respectively, to positive and negative values). The red dots represent the HCPs.

**Figure 4 molecules-27-04592-f004:**
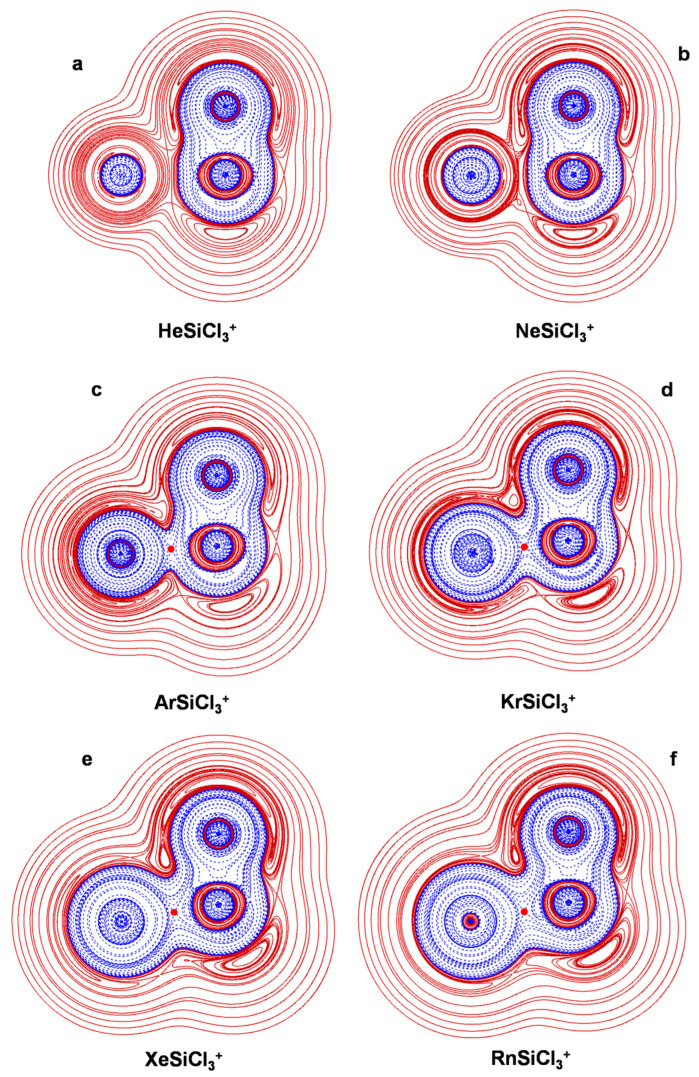
2D plots of *H*(***r***) in the symmetry plane of (**a**) HeSiCl_3_^+^; (**b**) NeSiCl_3_^+^; (**c**) ArSiCl_3_^+^; (**d**) KrSiCl_3_^+^; (**e**) XeSiCl_3_^+^; (**f**) RnSiCl_3_^+^ (solid/brown and dashed/blue lines correspond, respectively, to positive and negative values). The red dots represent the HCPs.

**Figure 5 molecules-27-04592-f005:**
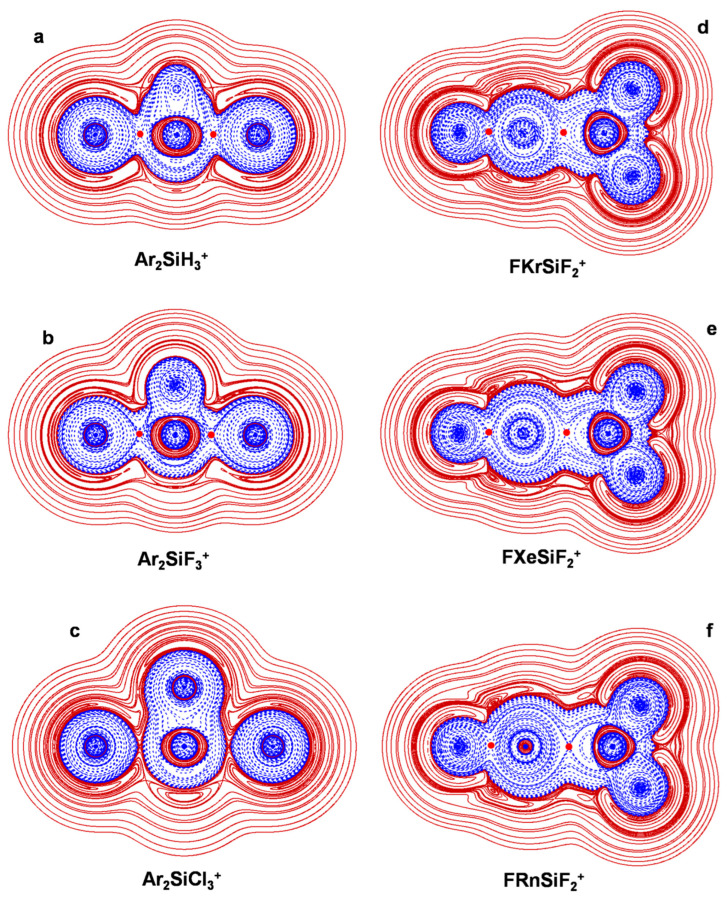
2D plots of *H*(***r***) in the symmetry plane of (**a**) Ar_2_SiH_3_^+^; (**b**) Ar_2_SiF_3_^+^; (**c**) Ar_2_SiCl_3_^+^; (**d**) FKrSiF_2_^+^; (**e**) FXeSiF_2_^+^; (**f**) FRnSiF_2_^+^ (solid/brown and dashed/blue lines correspond, respectively, to positive and negative values). The red dots represent the HCPs.

**Figure 6 molecules-27-04592-f006:**
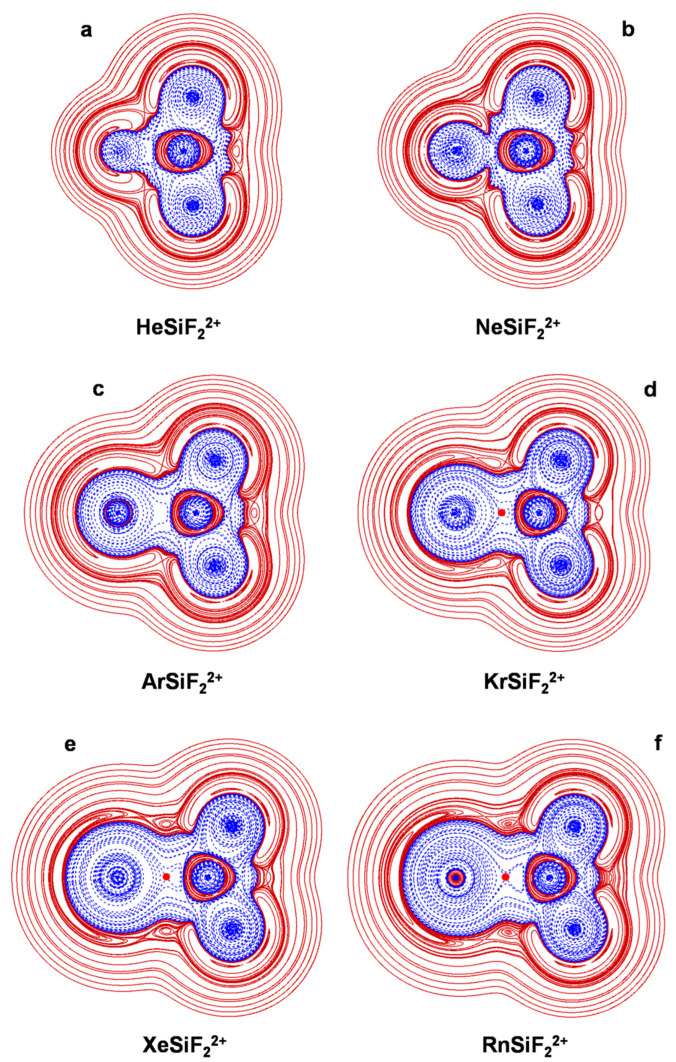
2D plots of *H*(***r***) in the symmetry plane of (**a**) HeSiF_2_^2+^; (**b**) NeSiF_2_^2+^; (**c**) ArSiF_2_^2+^; (**d**) KrSiF_2_^2+^; (**e**) XeSiF_2_^2+^; (**f**) RnSiF_2_^2+^ (solid/brown and dashed/blue lines correspond, respectively, to positive and negative values). The red dots represent the HCPs.

**Table 1 molecules-27-04592-t001:** Criteria to assign the Ng-X bonds of type A, B, or C in terms of the sign of the *H*(***r***) at around the BCP.

	*H*(*r*) at Around the BCP	
Bond type	Ng side	X side
A	negative	negative
B^l^ or C^l^	positive	positive
B^t^ or C^t^	positive	negative
	negative	positive
	negative	negative

**Table 2 molecules-27-04592-t002:** Criteria to assign the Ng-X bonds in terms of covalency.

	*ρ_s_*(ave) ^a^	*H*(*Ω_s_*)	Notation
Cov	≥0.08	invariably negative	H^−^
pCov	<0.08	invariably negative	H^−^
	any value	from negative to positive	H^+/−^ (positive on the average)
			H^−/+^ (negative on the average)
nCov	any value	invariably positive	H^+^

^a^*ea*_0_^−3^.

**Table 3 molecules-27-04592-t003:** MP2/aVTZ type and properties of the Ng-Si bonds of the Ng-Si cations. *R* (Å) is the bond distance; *D*_e_ is the electronic dissociation energy (kcal mol^−1^); *Ω_s_* is the volume (*a*_0_^3^) enclosed by the *s*(***r***) = 0.3 RDG isosurface at around the BCP; and *N*(*Ω_s_*), *ρ*_s_(ave), *H_s_*(ave/max/min) and BD_s_(ave) are, respectively, the total electronic charge (m*e*), the average electron density (*e a*_0_^−3^), the average, maximum and minimum value of *H*(***r***) (hartree *a*_0_^−3^), and the average BD (hartree *e*^−1^) over *Ω_s_*.

	Bond	*R*	*D* _e_	Type	*Ω_s_*	*N*(*Ω_s_*)	*ρ*_s_(ave)	*H_s_*(ave/max/min)	*H*(*Ω_s_*) ^a^	BD*_s_*(ave)	Assignment
HeSiH_3_^+^	He-Si	2.122	2.0	C^t^	0.3120	4.86	0.0156	−0.000075/0.00067/−0.00097	H^−/+^	0.0042	pCov(C^t^/H^−/+^)
NeSiH_3_^+^	Ne-Si	2.330	3.1	C^t^	0.4510	7.57	0.0168	−0.0013/−0.00008/−0.0025	H^−^	0.0747	pCov(C^t^/H^−^)
ArSiH_3_^+^	Ar-Si	2.406	14.1	A	0.9603	31.2	0.0325	−0.0092/−0.0043/−0.0123	H^−^	0.283	pCov(A/H^−^)
KrSiH_3_^+^	Kr-Si	2.514	19.3	A	1.2968	47.0	0.0363	−0.0124/−0.0063/−0.0167	H^−^	0.341	pCov(A/H^−^)
XeSiH_3_^+^	Xe-Si	2.664	26.5	A	1.9199	78.1	0.0407	−0.0156/−0.0082/−0.0215	H^−^	0.382	pCov(A/H^−^)
RnSiH_3_^+^	Rn-Si	2.736	30.1	A	2.2688	95.6	0.0421	−0.0164/−0.0089/−0.0233	H^−^	0.389	pCov(A/H^−^)
HeSiF_3_^+^	He-Si	2.060	2.5	A	0.3645	6.35	0.0174	−0.00075/0.00028/−0.0019	H^−/+^	0.043	pCov(A/H^−/+^)
NeSiF_3_^+^	Ne-Si	2.188	4.7	C^t^	0.5476	11.7	0.0215	−0.0029/−0.00082/−0.0049	H^−^	0.136	pCov(C^t^/H^−^)
ArSiF_3_^+^	Ar-Si	2.297	21.7	B^t^	0.7289	29.0	0.0398	−0.0142/−0.0064/−0.0220	H^−^	0.354	pCov(B^t^/H^−^)
KrSiF_3_^+^	Kr-Si	2.415	29.3	A	0.9947	45.0	0.0452	−0.0196/−0.0101/−0.0275	H^−^	0.431	pCov(A/H^−^)
XeSiF_3_^+^	Xe-Si	2.576	39.6	A	1.6579	88.6	0.0534	−0.0249/−0.0145/−0.0393	H^−^	0.470	pCov(A/H^−^)
RnSiF_3_^+^	Rn-Si	2.652	44.9	A	2.0403	113.0	0.0556	−0.0259/−0.0170/−0.0392	H^−^	0.471	pCov(A/H^−^)
HeSiCl_3_^+^	He-Si	2.986	0.4	C^l^	0.0880	0.30	0.0035	0.00095/0.0011/0.00087	H^−^	−0.272	nCov(C^l^)
NeSiCl_3_^+^	Ne-Si	2.916	1.0	C^l^	0.2278	1.53	0.0067	0.00057/0.00075/0.00042	H^−^	−0.086	nCov(C^l^)
ArSiCl_3_^+^	Ar-Si	2.690	6.1	A	0.9792	21.6	0.0220	−0.0035/−0.00030/−0.0074	H^−^	0.157	pCov(A/H^−^)
KrSiCl_3_^+^	Kr-Si	2.670	11.2	A	1.4951	46.0	0.0308	−0.0092/−0.0033/−0.0133	H^−^	0.295	pCov(A/H^−^)
XeSiCl_3_^+^	Xe-Si	2.746	19.3	A	2.1271	83.0	0.0390	−0.0146/−0.0072/−0.0215	H^−^	0.373	pCov(A/H^−^)
RnSiCl_3_^+^	Rn-Si	2.793	23.9	A	2.4402	102.9	0.0422	−0.0165/−0.0090/−0.0252	H^−^	0.391	pCov(A/H^−^)
Ar_2_SiH_3_^+^	Ar-Si	2.570	8.1	A	0.9572	23.8	0.0248	−0.0048/−0.0014/−0.0075	H^−^	0.190	pCov(A/H^−^)
Ar_2_SiF_3_^+^	Ar-Si	2.491	10.6	A	1.0771	30.9	0.0287	−0.0077/−0.0013/−0.0108	H^−^	0.265	pCov(A/H^−^)
Ar_2_SiCl_3_^+^	Ar-Si	2.961	3.9	C^t^	0.7456	9.96	0.0134	−0.000078/0.0012/−0.0014	H^−/+^	0.0036	pCov(C^t^/H^−/+^)
HeSiF_2_^2+^	He-Si	1.721	14.0	B^t^	0.1253	4.8	0.0383	−0.0057/0.0037/−0.0094	H^−/+^	0.155	pCov(B^t^/H^−/+^)
NeSiF_2_^2+^	Ne-Si	1.888	23.2	B^t^	0.1466	6.36	0.0434	−0.0065/0.0034/−0.0116	H^−/+^	0.159	pCov(B^t^/H^−/+^)
ArSiF_2_^2+^	Ar-Si	2.112	73.0	B^t^	0.3182	23.6	0.0743	−0.0374/−0.0165/−0.0541	H^−^	0.500	pCov(B^t^/H^−^)
KrSiF_2_^2+^	Kr-Si	2.247	94.3	A	0.9095	75.5	0.0830	−0.0455/−0.0268/−0.0791	H^−^	0.552	Cov(A)
XeSiF_2_^2+^	Xe-Si	2.428	123.2	A	2.4498	213.3	0.0871	−0.0477/−0.0287/−0.1095	H^−^	0.545	Cov(A)
RnSiF_2_^2+^	Rn-Si	2.518	137.4	A	2.7699	217.7	0.0786	−0.0403/−0.0239/−0.0660	H^−^	0.515	pCov(A/H^−^)
FKrSiF_2_^+^	Kr-F	1.927		A	0.3675	44.0	0.1197	−0.0476/−0.0264/−0.0973	H^−^	0.392	Cov(A)
	Kr-Si	2.477		A	2.1834	156.5	0.0717	−0.0356/−0.0158/−0.0656	H^−^	0.487	pCov(A/H^−^)
FXeSiF_2_^+^	Xe-F	1.983		A	0.5258	61.2	0.1164	−0.0553/−0.0337/−0.0825	H^−^	0.471	Cov(A)
	Xe-Si	2.631		A	3.1721	223.3	0.0704	−0.0339/−0.0132/−0.0707	H^−^	0.464	pCov(A/H^−^)
FRnSiF_2_^+^	Rn-F	2.054		A	0.4141	44.3	0.1071	−0.0403/−0.0254/−0.0591	H^−^	0.373	Cov(A)
	Rn-Si	2.735		A	3.2828	215.2	0.0656	−0.0307/−0.0104/−0.0701	H^−^	0.441	pCov(A/H^−^)

^a^ Depending on the sign of *H_s_*(ave/max/min), *H*(*Ω_s_*) = H^+^, H^+/^^−^, H^−/+^, or H^−^.

## Data Availability

Not applicable.

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
