# Peer review of "Noble Gas—Silicon Cations: Theoretical Insights into the Nature of the Bond"

_molecules, 2022, doi:10.3390/molecules27144592_

Round 1

Reviewer 1 Report

Borocci S. at  al. explored bonding motifs  for NgSiX3+ (Ng = He-Rn; X = H, F, Cl) and NgSiF22+ (Ng = He-Rn), the di-coordinated Ar2SiX3+ (X = H, F, Cl), and FNgSiF2+ (Ng = Kr, Xe, Rn). Obtained data were compared with previous findings already available in the literature. The conclusion reached with previous study ref. 22. However, the article cannot publish in current form. The main problems of the study:

1. The authors didn’t formulate what the study purpose and why they want to repeat the same studies. Moreover, according the authors, they were achieved in 2013 already. This explanation and literature review I would expect in the introduction.

2. It is unclear what information provide Figures 2-4 as they are the pure data. I suggest to move it to supplementary material and provide one Figure with detailed explanations about the importance or details the reader should focus.

3. Finally, it is unclear what new information the authors gathered. They even didn‘t try to explain or discuss the important of the explored data and what is the main result.

Author Response

Borocci S. at  al. explored bonding motifs  for NgSiX3+ (Ng = He-Rn; X = H, F, Cl) and NgSiF22+ (Ng = He-Rn), the di-coordinated Ar2SiX3+ (X = H, F, Cl), and FNgSiF2+ (Ng = Kr, Xe, Rn). Obtained data were compared with previous findings already available in the literature. The conclusion reached with previous study ref. 22. However, the article cannot publish in current form. The main problems of the study:

  1. The authors didn’t formulate what the study purpose and why they want to repeat the same studies. Moreover, according the authors, they were achieved in 2013 already. This explanation and literature review I would expect in the introduction.

Response

The Reviewer focuses attention on the species already investigated in 2013 and accounted in Ref. 22 (Ref. 27 of the revised manuscript), namely the NgSiX3+ (Ng = He-Rn; X = H, F, Cl) and Ar2SiX3+ (X = H, F, Cl). As a matter of fact, we extended also the study to the NgSiF22+ (Ng = He-Rn) and FNgSiF2+ (Ng = Kr, Xe, Rn), not investigated in Ref. 27, and whose bonding situation is still unexplored. The main purpose of this study was, in fact, to perform a comparative theoretical analysis of noble gas-silicon cations, with emphasis on the relationships between the various experimentally observed bonding motifs, and the nature of the Ng-Si bonds. The NgSiX3+ (Ng = He-Rn; X = H, F, Cl) and Ar2SiX3+ (X = H, F, Cl) were recalculated in order to have data strictly comparable with those obtained for the NgSiF22+ (Ng = He-Rn) and FNgSiF2+ (Ng = Kr, Xe, Rn). All these considerations are given in the second yellow-highlighted paragraph of the Introduction, and in the two sentences at p. 14.

  1. It is unclear what information provide Figures 2-4 as they are the pure data. I suggest to move it to supplementary material and provide one Figure with detailed explanations about the importance or details the reader should focus.

Response

Figures 2-4 are indeed of major interest to discuss the bonding situation of the NgSiX3+ (Ng = He-Rn; X = H, F, Cl), and it is not easy to replace them with a single Figure containing the relevant information. In addition, for consistency, we should also cut Figures 5 and 6, which illustrate the bonding situation of the Ar2SiX3+ (X = H, F, Cl), FNgSiF2+ (Ng = Kr, Xe, Rn), and NgSiF22+ (Ng = He-Rn). Rather, based on the Reviewer’s comment, we added a paragraph (the yellow-highlighted one at p. 6) explicitly mentioning the major interest of these Figures and their provided information.

  1. Finally, it is unclear what new information the authors gathered. They even didn‘t try to explain or discuss the important of the explored data and what is the main result.

Response

The information gathered in this study and the main obtained result was a comprehensive view of the nature of the bonds occurring in the various experimentally-observed noble gas-silicon cations, and in their still unreported congeners. This is explicitly mentioned in the yellow-highlighted paragraph added to the Concluding Remarks.

Reviewer 2 Report

This manuscript discussed the bonding features of a series of noble gas-silicon compounds (mono-coordinated, di-coordinated and inserted). This is a thorough and high-quality research report which has specific research target and clear outlines. The authors not only provided basic analysis data but also comprehensive induction of the bonding tendency. This manuscript can be published on Molecules after minor reversion. My detailed suggestions are listed below:

Suggestion 1:

In introduction

Could you please add more background about “Noble Gas–Silicon Cations”? Why did you research the “Noble Gas–Silicon Cations”? Are they interstellar compounds? Or are they have potential industrial usage? Noble Gas can form bonds with many elements such as Ge, Sn, Pb……

Suggestion 2:

Bond Dissociation Energies (Table 1)

In my opinion, the most important index, Bond Dissociation Energies, is missed in your manuscript. It’s the most direct way to measure the strength of the bonds. Could you please add these data into table 1.

Suggestion 3:

Experimental data

Ab initio results is very accurate for gas phase compounds. If you can provide some comparison between experimental data (like IR spectroscopy) and your ab initio simulation results, that will provide solid supports to you conclusion.

Comment 1:

Relativistic effective

In Page 4, Line 167, “… scalar relativistic …”. Ar, Kr, Xe, Rn elements have strong relativistic effects. And this can affect some key properties such as reaction rate(https://doi.org/10.1016/j.molstruc.2021.131572.). Is it necessary to locate the relativistic effects in bonding features?

Author Response

This manuscript discussed the bonding features of a series of noble gas-silicon compounds (mono-coordinated, di-coordinated and inserted). This is a thorough and high-quality research report which has specific research target and clear outlines. The authors not only provided basic analysis data but also comprehensive induction of the bonding tendency. This manuscript can be published on Molecules after minor reversion. My detailed suggestions are listed below:

Response

We thank the Reviewer for his appreciation, and addressed his/her suggestions and comment as detailed below:

Suggestion 1:

In introduction

Could you please add more background about “Noble Gas–Silicon Cations”? Why did you research the “Noble Gas–Silicon Cations”? Are they interstellar compounds? Or are they have potential industrial usage? Noble Gas can form bonds with many elements such as Ge, Sn, Pb……

Response

The first yellow-highlighted paragraph added to the Introduction mentions also the reasons of interest of noble gas-silicon cations.

Suggestion 2:

Bond Dissociation Energies (Table 1)

In my opinion, the most important index, Bond Dissociation Energies, is missed in your manuscript. It’s the most direct way to measure the strength of the bonds. Could you please add these data into table 1.

Response

Bond Dissociation Energies (De) are quoted in the fourth column of Table 1 and discussed in the text.

Suggestion 3:

Experimental data

Ab initio results is very accurate for gas phase compounds. If you can provide some comparison between experimental data (like IR spectroscopy) and your ab initio simulation results, that will provide solid supports to you conclusion.

Response

Unfortunatley, there are no available experimental data such as those mentioned by the Reviewer.

Comment 1:

Relativistic effective

In Page 4, Line 167, “… scalar relativistic …”. Ar, Kr, Xe, Rn elements have strong relativistic effects. And this can affect some key properties such as reaction rate(https://doi.org/10.1016/j.molstruc.2021.131572.). Is it necessary to locate the relativistic effects in bonding features?

Response

The long-standing experience with the use of pseudopotentials for the heaviest noble gases clearly demonstrates that it is not necessary to locate the relativistic effects in bonding features.

Round 2

Reviewer 1 Report

I think the authors answered the raised problems and it can be published as there are some new aspects.